# Computational Modelling for Electrical Impedance Spectroscopy-Based Diagnosis of Oral Potential Malignant Disorders (OPMD)

**DOI:** 10.3390/s22155913

**Published:** 2022-08-08

**Authors:** James P. Heath, Keith D. Hunter, Craig Murdoch, Dawn C. Walker

**Affiliations:** 1Department of Computer Science, University of Sheffield, Sheffield S1 4DP, UK; 2Liverpool Head and Neck Centre, Molecular and Clinical Cancer Medicine, University of Liverpool, Liverpool L69 7TX, UK; 3School of Clinical Dentistry, University of Sheffield, Sheffield S10 2TA, UK

**Keywords:** impedance spectroscopy, oral cancer, oral potential malignant disorder, finite element modelling, histology

## Abstract

A multiscale modelling approach has been applied to the simulation of the electrical properties of oral tissue, for the purpose of informing an electrical impedance-based method of oral potential malignant disorder (OPMD) diagnosis. Finite element models of individual cell types, with geometry informed by histological analysis of human oral tissue (normal, hyperplastic and dysplastic), were generated and simulated to obtain electrical parameters. These were then used in a histology-informed tissue scale model, including the electrode geometry of the ZedScan tetrapolar impedance-measurement device. The simulations offer insight into the feasibility of distinguishing moderate dysplasia from severe dysplasia or healthy tissue. For some oral sites, simulated spectra agreed with real measurements previously collected using ZedScan. However, similarities between simulated spectra for dysplastic, keratinised and non-dysplastic but hyperkeratinised tissue suggest that significant keratinisation could cause some OPMD tissues to exhibit larger than expected impedance values. This could lead to misidentification of OPMD spectra as healthy. Sources of uncertainty within the models were identified and potential remedies proposed.

## 1. Introduction

### 1.1. Objectives

Currently, the diagnosis of oral potentially malignant disorders (OPMD) relies on clinical examination and histological analysis of tissue biopsies [1]. Taking biopsies is painful for the patient, processing to produce the histology slide is time consuming and the histological analysis requires significant training. The result of the histological analysis is qualitative, with sub-optimal sensitivity and specificity and with issues of poor reproducibility between pathologists [2]. Longitudinal monitoring of potentially malignant sites by repeat biopsy is not currently a possibility many patients will tolerate. By contrast, using electrical impedance spectroscopy (EIS) to measure the electrical response of clinically abnormal oral tissue, a clinician can obtain a real-time, quantitative measure of tissue health with minimal distress to the patient. By using simulations, it is possible not only to understand the difference in electrical characteristics between OPMD and healthy tissue, but also explore the feasibility in monitoring the progression of premalignant to oral cancer: specifically, if tissue structures associated with increasing severity of dysplasia are simulated, how would this result in changes in the measured impedance spectra, and are they sufficiently large to be discriminated?

In this paper, we describe the simulation of impedance spectra of healthy, benign hyperplasia and potentially malignant (OPMD) oral tissue using the finite element method. To ensure the realism of the simulation, the models are informed by histological images of oral tissue (i.e., capturing the geometry of tissue and its constituent cells). At tissue level, simulation also includes the specifications of the ZedScan probe (tetrapolar electrode configuration, operating current and frequency range). The results obtained can inform the development of EIS measurement devices for clinical applications by providing spectra templates of different pathologies and suggesting potential future improvements to the measurement hardware.

### 1.2. Background: Electrical Impedance Spectroscopy as a Diagnostic Tool

Electrical impedance spectroscopy (EIS) is a technique based on the measurement of the frequency-dependent electrical response (or impedance) when a small alternating current (AC) is passed through a material. If the frequency of the AC signal is varied, it is possible to detect the response of multiple components within a material, depending on their capacitance and resistance [3]. In the context of biological tissue, the capacitive nature of cell membranes leads to current flow that is frequency-dependent, with current confined to limited extracellular volume at low frequencies, but able to penetrate the interior of cells at higher frequencies, leading to a characteristic beta dispersion [4], or significant fall in measured impedance in the 10–100 kHz region. Given that the polarisation of cell membranes is critical in this behaviour, cell damage that leads to the disruption of membranes, or physiological changes that lead to alterations in the size, shape or cohesion of cells, can be detected by the measurement and analysis of impedance spectra [5]. Electrical impedance spectroscopy can thus be applied as a diagnostic tool. In this context, parameterised EIS measurements have been used to detect inflammation in oral mucosa [6] for the diagnosis of Barrett’s oesophagus in oesophageal tissue [7], and the diagnosis of cervical intraepithelial neoplasia in the human uterine cervix [8,9]. In addition, it has been applied to the detection/diagnosis of skin malignancies [10,11], changes in the prostate [12], bladder pathology [13] and as a predictor of pre-term labour [14].

In a previous study, EIS measurements of cancerous, OPMD and healthy oral tissues were collected using the ZedScan device, as illustrated in Figure 1 [15]. The data from this study showed a reduction in the real impedance for cancerous tissue at frequencies less than ~100 kHz when compared to normal oral mucosa (Figure 2). However, there was variation in results according to the type of oral tissue from which data were collected, and even within the same site in the mouth. The current study was prompted by the need to further understand the causes underlying the variation in results and potentially improve future versions of ZedScan to optimise the potential for differentiation between normal, OPMD and cancerous oral tissues.

### 1.3. Computational Modelling of EIS

Computational modelling can be employed to evaluate and quantify the specific contributions of the various aspects of the tissue structure of interest to characteristics of the measured impedance curves. This adds further value to the process of simply collecting and sorting EIS data according to subsequent diagnosis/categorisation of the tissue site in question. Models provide a platform in which “inputs”, e.g., the size and arrangements of cells, can be altered as the sensitivity of the subsequent “outputs”—impedivity at a range of frequencies—can be assessed.

For this purpose, Walker et al. developed multiscale finite element simulations of EIS of cervical [16] and bladder [17] tissue, in order to better understand EIS measurements on these tissue types recorded using a precursor to the ZedScan device. The cervical model simulations show a characteristic drop in impedance for dysplastic tissue, in good agreement with measured data. Simulation studies focused on the bladder offered potential explanations for some of the more complex behaviour seen in this tissue, whereby an increase in impedance was routinely recorded from dysplastic, compared to normal, sites. Both studies highlighted the role of the morphology and packing of the cell-dense epithelia in determining the characteristics of the measured impedance spectra, as well as the role of the underlying connective tissue and surface fluids. More recently, a finite element approach was used by Huclova et al. [18,19] in order to study the dielectric properties of human skin. These investigators also employed a multiscale approach, whereby the dielectric tensors of different cell types were calculated, then these used as material properties for a tissue scale model. Results suggest that the dielectric properties in the 0.1–1 MHz region are sensitive to cell shape.

A multiscale finite element modelling approach is employed in this study, where we aim to explore the factors underlying the differences in characteristic spectra associated with different oral tissue types.

### 1.4. Oral Tissue Structure and EIS Measurements

The outermost layer of oral tissue, the oral epithelium, consists of distinct strata of epithelial cell types, as shown in Figure 1a. The deepest are basal cells: progenitor cells that divide to make new cells for the epithelium. After division, the cells migrate towards the top of the epithelium, and flatten. Initially, the cells at intermediate depth become prickle cells and finally superficial cells. Superficial cells associated with some oral sites become increasingly keratinised as they reach the top of the epithelium, providing increased wear resistance when required. Examples of non-keratinised tissue are the buccal and labial mucosa, while keratinised sites include the hard palate and gingiva. Underlying the basal cells, there is a layer of connective tissue, or lamina propria, that contains fibroblasts and blood vessels and at varying depths, again according to site, bone, fat or muscle. The histological changes in oral tissues which are related to an increased risk of the development of oral cancer are termed oral epithelial dysplasia. Many changes in the overall architecture of the epithelium and of individual cells contribute to this assessment of the epithelium by a pathologist.

Previous modelling work studied the sensitivity of surface impedance measurements to epithelial cell morphology [16,20], suggesting that the local impedance increases closer to the surface of the epithelium in healthy tissue, where superficial cells are highly flattened with minimal extracellular space. By contrast, dysplasia is associated with a reduction in differentiation, with cells exhibiting a morphology more typical of basal cells, with higher extracellular space volumes, leading to reduced local impedance. A computational model of epithelial tissue structure must therefore account for the depth-related distributions in both epithelial cell morphology, and extracellular volume. In addition, though most current flows through the epithelial layers and underlying connective tissue in some oral sites, it may be necessary to consider underlying layers, including muscle (tongue), bone (hard palate) and fat (buccal mucosa) that is abundant in some oral tissues.

Specific to the mouth, hyperkeratinised tissues must also be considered. At certain sites this is normal, but hyperkeratinisation of oral tissues occurs where environmental factors, such as trauma, cause an increase in epithelial keratinisation as a response. It is possible for this keratinisation to occur in the absence of epithelial dysplasia; hence it is important to understand if there is a difference in impedance response of such tissue with and without dysplasia, so this condition is not misidentified.

The aim of the modelling study described here is to extend computational modelling methodologies previously applied to other epithelial tissues, in order to fully understand the electrical characteristics of the much more varied (in terms of thickness and keratinisation) tissues found in the mouth. The long-term goal is that this knowledge will aid the interpretation of EIS spectra and inform the feasibility of applying this measurement technique for the discrimination of oral tissue pathologies.

## 2. Materials and Methods

### 2.1. Histology Analysis

In order to produce realistic simulations of oral tissue, the finite element models must have equivalent geometries to real oral tissue. To obtain morphological parameters, manual measurements of digitally scanned histological images (see Figure 1a) of several oral sites with varying pathologies (normal, severe and moderate dysplasia) and a varying degree of keratinisation were taken using Imagescope software (Leica-Aperio Ltd., Nussloch, Germany). Approximately 20–40 individual cell length and area measurements were made for each of the three cell compartments (basal, prickle cell and superficial). Cell size measurements are included in Appendix A, Appendix A.

The thickness of each cellular compartment and the keratinised layer (where present) were also recorded and the average values calculated, along with the length and breadth of the rete processes. As the basal layer is typically only one or two cells thick, the maximum cellular dimension was used for the tissue scale basal layer thickness. As an approximation, the rete peg length was used as the prickle layer thickness and the superficial layer thickness was assumed as the remaining depth after the basal and prickle layers were subtracted from the total thickness. The mean values obtained were used to construct the baseline finite element models (see below) and upper and lower bounds (see Table 1) were later simulated by using the mean value ± one standard deviation for all values.

### 2.2. Finite Element Modelling

#### 2.2.1. General Application of Finite Element Modelling (FEM) to EIS

The finite element technique is a numerical analysis method commonly used in the solution of physics-based field problems. In the case of AC electrical problems, the objective is to calculate an approximate solution for the potential distribution within a volume partitioned into elements, connected at nodes (i.e., a mesh) in order to solve the following equation:(1)$$\left[K_{\sigma }\right]\left\{\phi \right\}+\left[K_{\varepsilon }\right]\left\{\dot{\phi }\right\}=\left\{I\right\}$$
where *ϕ* is the vector of nodal voltages, *I* is the vector of nodal currents, *K_σ_* and *K_ε_* are matrices of internal conductivities and permittivities, respectively. These are sometimes referred to as the global stiffness matrices due to FEM’s original application to mechanical problems. Further information relating to this technique and applications to electrical biomedical problems can be found in [21].

In order to simulate EIS, it is necessary to create a mesh representing the geometries of the real-world tissue structures of interest and assign electrical material properties (conductivity and permittivity) to each element to model the conductive and capacitive behaviour of tissue, respectively. Imposed boundary conditions can include a known potential difference applied across a model or a known applied current. Any surface that does not have a set value for potential or current has its current density set to zero to confine the current within the model.

Following the solution of Equation (1), if both the nodal potentials and the currents are known, then transfer impedance between any subset of nodes can be calculated directly simply by applying Ohm’s law at each frequency.
Z* = V/I*(2)

As magnetic effects are not of interest, induction is neglected. Other key assumptions are that material properties with respect to the electric field are time-invariant and linear.

#### 2.2.2. Multiscale Approach

As described above, cellular membranes are critical in determining the characteristic beta dispersion associated with cellular tissues, so should be represented in the FEM mesh. This presents a challenge, as a cell membrane is extremely thin (of the order of 8 nm) compared to the cytoplasm (typically several μm), so to model a cell with distinct material properties for the membrane and cytoplasm, many small elements (and hence many nodes) are required. This impacts on the computational cost of the simulation (as each node is effectively represented by an equation in the FEM method), and it is not computationally tractable to simulate the volume of tissue equivalent to a clinical measurement with cellular scale detail.

The solution adopted is to use a multiscale modelling approach, whereby the different cell types are simulated with a high level of detail, and their electrical properties calculated using the FEM approach described above. The effective conductivity and permittivity at each frequency of interest are then used as material properties in appropriate elements in a second, tissue scale FEM model, allowing for cellular scale information to be passed to a larger length scale. The frequency range modelled spans the range between 76 Hz and 625 kHz, with the specific frequencies reflecting those used by the ZedScan probe (as listed in the Appendix A). The multiscale modelling process, similar to that applied previously for cervix [16] and bladder [17] is summarised in Figure 3 and detailed in the following sections. For clarity, the two models are referred to as the *cellular scale model* and the *tissue scale model*.

Both cellular and tissue scale simulations were implemented using a commercial finite element modelling tool ANSYS APDL [22]. Specific modules used are listed in the Appendix A (Appendix A) with appropriate mesh size convergence and size effect studies for both scales considered (Appendix A).

#### 2.2.3. The Cellular Scale Model

The generic model for a cell consists of a nested cube structure, which can have a spherical nucleus in its centre, if the cell is nucleated. The outer layer is designated as extracellular space (ECS), the next layer in is the cell membrane and finally there is a cubic volume of cytoplasm. For nucleated basal cells, the nucleus is represented by a sphere (nuclear interior, or karyoplasm) surrounded by a spherical shell (nuclear membrane) as shown in cross-section in Figure 4a. The dimensions of the cell can be manipulated in the XYZ directions to produce cells with different aspect ratios, but the nucleus remains a sphere with a fixed radius (2 μm). Membrane thicknesses are constant for all cell aspect ratios. Nuclear morphology is known to change during dysplasia; however, simulations of the cervix showed that the nucleus has negligible effect on the electrical response [20]. Nuclei are not generally apparent in superficial cells in histology sections, so these cells are modelled as anuclear.

Basal, prickle and superficial cell models were generated based on histology images as described above, with a list of all cellular scale dimensions, as obtained from the histology analysis described above, given in Appendix A Appendix A. No literature values for dimensions of ECS in oral tissue due to dysplasia exist, so ECS thicknesses were assumed to be the same as those previously used for modelling in normal and dysplastic cervical epithelia [16,20]—see Appendix A Appendix A. Here, it is assumed that cervical precancer grade CINII and CINIII are equivalent to moderate and severe dysplasia in oral tissue, respectively. Properties assigned to the various cellular compartments are shown in the Appendix A (Appendix A).

To calculate the impedance response of a group of cells, a composite model was created by tessellating individual cell models with a 0.5 x cell offset in the XY plane (i.e., parallel to the tissue surface) to create a “bricked” arrangement (Figure 4b) and selecting a subsection of the model to represent the smallest repeating unit (Figure 4c). A consequence of this “bricked” arrangement is the total impedance of a collection of brick layered cells is different in the Z direction (Figure 4d) compared to the X or Y directions (Figure 4e), the latter two being equivalent. Hence, for every cell type both Z and XY properties must be calculated. For the latter case, this is done by the application of an arbitrary potential difference of 1 V across the model by setting the potential to 1 V for coupled sets of nodes on the top of the cells and 0 V at the bottom (Figure 4f). As we were using harmonic analysis, the 1 V potential difference was applied as a sinusoid with 1 V peak voltage. All other free surfaces of the cell had an implicit Neumann condition, setting the current density to zero. This confined the current to flow only through the cell. Another consequence of the harmonic analysis was that the current value obtained was a complex number, allowing the impedance to be easily calculated from Equation (2). This process was repeated for each frequency required. Effective properties were calculated for the X–Y directions by rotating the boundary conditions through 90 degrees and repeating the process.

Once the impedance of a cell type unit was calculated in both directions, the impedance was then converted into a size-independent impedivity (ρ*) as follows:ρ* = (Z*A)/l(3)
where A is the area normal to the applied voltage and l is the length parallel to the applied voltage. A convergence study with increasing numbers of symmetry units was carried out and is presented in the Appendix A, Appendix A.

Impedivity was then converted into effective resistivity and relative permittivity at each frequency in the XY and Z directions. Resistivity is simply the real component of impedivity. The relative permittivity was calculated from the imaginary component of the cellular scale impedance (Z″) as follows: 

Imaginary admittance component Y″ was calculated as
Y″ = Z″/(Z′^2^ + Z″^2^)(4)

Capacitance C was given by:C′ = Y″/ω(5)
and relative permittivity
ε_r_ = (C′l)/(ε_0_A).(6)
where ε_0_ is the permittivity of free space, l the effective length of the block simulated, A the cross-sectional area and C is obtained from the imaginary admittance component. This process was repeated for each simulated frequency corresponding to the frequencies used by the measurement probe. Once the resistivity and relative permittivity had been obtained for each cell type and for both direction, these were then input into the tissue scale model. The transfer of information between the cellular and tissue scale model is summarised in Figure 3.

#### 2.2.4. The Tissue Scale Model

The tissue scale model consisted of a cuboid divided into as many layers as required: one for each of the three epithelial tissue layers, a stroma layer and surface layers including saliva and keratinised tissue (see Figure 5). The area of tissue simulated was 40 mm by 40 mm, as this was found to be large enough to avoid boundary effects for a similar probe design [20]. A total tissue depth of 5 mm was found to be adequate (see Appendix A Appendix A), also in good agreement with [20]. Tissue layer depths obtained from histology are shown in Table 1.

The same principles for simulating EIS at the cellular scale were applied at the tissue scale, with the addition of the definition of boundary conditions to represent the tetrapolar electrode array, as shown in Figure 6. Specifically, the electrodes were modelled as circles attached to the outer surface of the top layer in the tissue scale model (see Figure 6a). The nodes of each of the electrode areas were defined as unique coupled sets. The electrode spacing and radii were taken from the specifications of the ZedScan device (see Figure 6b). The drive electrode had an applied current of 6.08 μA (AC peak current, as per the ZedScan device) and the receive electrode was grounded, by setting the potential to 0 V.

At each simulated frequency, the current flowing out of the grounded electrode was calculated. The remaining two electrodes are referred to as the measurement electrodes, and the potential difference between each was measured for the calculation of impedance at a given frequency using Equation (2), which could then be plotted to give the simulated frequency-dependent impedance for that tissue (see Appendix A).

Epithelial layer conductivities and permittivities were obtained from the cellular level model simulations, as described above. Electrical properties of the stroma were taken from the literature [20,23]. Given that the oral cavity is a hydrated environment, it is reasonable to assume that a thin conductive layer existed as an interface between the probe electrodes, and a saliva layer thickness of 0.01 mm was included (based upon a mucus layer previously used in equivalent simulations of the cervix) [20]. The saliva resistivity was set to 1 Ωm and relative permittivity was set to 72, based upon generic values of an isotropic, biological fluid [20].

No electrical properties of the keratinised layer in the oral cavity exist in the literature; hence these parameters were fitted by comparing several systems with plausibly similar characteristics, as described in the following section. Finally, the effects of explicit inclusion of the electrical properties of the underlying sub-stromal layers according to specific sites in the oral cavity were included.

## 3. Results

Given that the main focus of this study was the spectra derived from the tissue level model, for brevity, the cellular scale results are given in the Appendix A (see Appendix A). The trends in frequency-related impedance characteristics were in good agreement with the cervix simulations [20], with differences in magnitude attributable to the different cell sizes obtained from the oral tissue histology. It should be noted that while the cell types in Appendix A are listed by tissue type (e.g., degree of keratinisation) no keratinised layer was present at the cellular scale, hence any difference in impedivity shown in Appendix A is resultant from dysplasia (i.e., differences in cell morphology and ECS). The results presented below relate to those obtained using these cellular scale results as material properties in the relevant tissue scale model, as previously described.

A significant difference between tissues previously modelled (e.g., cervix, bladder) and many oral tissues is the presence of a surface keratin layer, and initial investigations focused on exploring the effect of this layer on the simulated impedance spectra. Given the lack of documented electrical properties for keratin in the oral cavity, this study was crucial in selecting the most appropriate properties to assign for further simulations.

### 3.1. Effect of Keratinised Layer Properties

#### 3.1.1. Frequency-Independent Keratin Properties

Initially, the keratinised layer was assigned the electrical properties of pure keratin [24]. Literature values show the resistivity reduces with increasing hydration [25]; in the context of oral tissue, this is likely due to a higher concentration of mobile ions, plus potentially wider extracellular routes available for conduction (the latter being important at low frequencies). To test the effect of hydrated keratin properties, keratinised normal and OPMD tissue, simulations using layer thicknesses from Table 1 (cases 2 and 4 respectively) were performed. The keratinised layer resistivity varied from 0.1 MΩm to 1 kΩm, with the upper bound from [24] and the lower bound chosen so that the keratinised layer would always be more resistive than the superficial cell layer. The relative permittivity was kept constant at 1.17 [24]. Properties were initially assumed not to vary with frequency.

Figure 7 shows that high-resistivity surface layers would result in high-impedance measurements at all frequencies and reducing the keratinised layer resistivity below 5 kΩm recovers the dispersion in the spectra above 10 kHz, as observed with the measurement probe (see Figure 2). If the keratinised layer resistivity was lowered to 1 kΩm, then the magnitude of the impedance at low frequency approached that observed in real measurements for normal and OPMD tissue, though simulated spectra plateaued to much higher frequencies.

#### 3.1.2. Frequency-Dependent Keratin Properties

Using the same keratinised layer thickness measurements as for the previous section (Table 1), additional spectra were simulated using frequency-dependent values of permittivity and resistivity (see Appendix A, Appendix A) derived from the top layer of human skin (stratum corneum), which is similar in structure to that of keratinised oral mucosa [26]. The low-frequency resistivity of the stratum corneum is less than keratin (0.049 MΩm at 76 Hz); at 0.625 MHz, the resistivity of this layer was reduced to 65 Ωm. The permittivity varied from 30 to 5.7 at 76 Hz and 0.625 MHz, respectively. All properties were assumed to be isotropic. The previous assumption that hydrated keratin is less resistive was applied to justify reducing the resistivity of the stratum corneum values for use in the mouth by up to a factor of forty, with this lower bound chosen so the keratinised layer remained more resistive than the underlying epithelium.

Reducing the resistivity of the keratinised layer, as described above, recovered the expected trend with pathology (see Figure 8 and Figure 2 for comparison). Frequency-dependent keratin properties, with the maximum (/40) adjustment to represent hydration, were then used in the simulations described below.

### 3.2. Effect of Dysplasia

Simulations of the specific normal, hyperkeratinised and OPMD tissue structures listed in Table 1 were carried out, in order to examine the changes in impedance spectra as a result of pathological change (or non-pathological change in the case of hyperkeratinisation). All models used the same basic structure, as shown in Figure 5, and each pathology was simulated by adjusting the thickness of each cellular compartment according to the histology data for the differing tissue type (as per Table 1). The calculated cellular level electrical properties for the relevant epithelial strata (shown in Appendix A) were assigned as appropriate. The resulting spectra are shown in Figure 9. In order to account for potential variation in cellular compartment thickness, upper and lower bounds for the spectra templates were calculated using the mean compartment thickness ± one standard deviation (Table 1) for all tissue types. These bounds are represented by error bars for each median thickness datapoint.

Visual comparison of Figure 2 and Figure 9 reveals several similarities between the simulated and measured data. Simulated normal tissue spectra exhibit relatively high impedance of ~4 kΩ at low frequency, plateauing, then falling to ~200 Ω at high frequency with a sharp dispersion in the ~10 kHz region, overlapping in characteristics with the majority of spectra measured at various tissue sites subsequently diagnosed as normal in Figure 2. By contrast, the simulated para-keratinous with severe OPMD cases have a significantly reduced low-frequency impedance, falling to similar values to the normal cases at high frequency, more representative of the tissues diagnosed as cancerous in Figure 2. Interestingly, the spectra associated with the hyperkeratinous (no OPMD), keratinous (moderate OPMD) and keratinous (OPMD) seem to form a third group falling between the latter two. These spectra all exhibit high low-frequency impedance (similar, or higher than the normal tissue cases), followed by a gradual impedance decrease with increasing frequency, rather than a sharp dispersion. There is significant overlap between the spectra for hyperkeratinous (no OPMD) and keratinous with severe OPMD spectra in particular. There is significantly more variation in the OPMD and hyperkeratinous spectra compared to normal cases.

In order to explore further how the presence of a keratinised layer could confound the impedance spectra from healthy and OPMD tissue, a systematic comparison of hyperkeratinised (non-dysplastic) and keratinised (OPMD) spectra was conducted by repeating these simulations with a variable surface keratinised layer thickness of 15 μm to 90 μm. Underlying cellular compartment thicknesses were taken from cases two and three (see Table 1) for the non-dysplastic and OPMD simulations, respectively. This range was based upon the minimum and maximum keratinised layer thickness values found by histological analysis (see Table 1). As shown in Figure 10, increasing the keratinised layer thickness for the OPMD models gradually increased the low-frequency impedance values towards the values obtained for healthy values. This shows our model was highly sensitive to keratinised layer thickness.

### 3.3. Effect of Underlying Tissue

In some oral sites, the assumption of up to 5 mm of stroma underneath the epithelium is not realistic. In the tongue (case 3) there is significant muscle tissue. For the buccal mucosa (cheek, case 1) there is a layer of fat. Finally, for the hard palate (case 2) there is bone close to the epithelium. For these three cases the stroma layer was reduced to 1 mm (based upon histology measurements) and the rest of the 5 mm of tissue was assigned frequency-dependent fat, bone or muscle properties, respectively [27].

For the buccal mucosa, hard palate and tongue, addition of these extra layers resulted in negligible changes to the spectra when compared to experimental uncertainty (see Figure 11).

## 4. Discussion

EIS is a tool that has been used clinically for the differentiation of normal and OPMD tissues in the cervix, and previous modelling studies have provided insights into how the spectra measured from tissue in the cervix, as well as other epithelial tissues such as bladder, can relate to aspects of the underlying tissue structure. Unlike cervix, oral tissues have additional complexities that do not have direct equivalents, most notably the presence of a surface keratinised layer that can be present at some oral cavity sites and in some pathologies (including non-dysplastic, hyperkeratinised tissue), and sub-mucosal layers including fat, bone and muscle. This study has attempted to explore the role of both these surface and sub-mucosal layers in influencing the electrical impedance that would be measured using a surface tetrapolar array.

A significant challenge in quantifying these effects using a modelling approach is the lack of material property measurements of the keratinised layer in the context of the oral cavity. Given the lack of information, the tissue scale model was initially fitted to the real spectra of healthy and OPMD oral tissue measured by [15] (following our initial sensitivity study varying the resistivity of the keratinised layer between 1 MΩm and 1 kΩm to reflect a decrease in resistivity due to hydration). It was apparent that the initial value of keratinised layer resistivity, based on dry keratin properties, was too high to allow penetration into the underlying tissue. Most of the current was trapped in the surface saliva layer, resulting in a complete lack of dispersion (Figure 7). This 1 MΩm value represented an extreme upper bound of the resistivity of the keratinised layer as (i) the oral keratinised layer is not pure keratin, but flattened cells with increased keratin content, and (ii) realistically, we would expect some degree of hydration of the keratinised layer, given the moist nature of the oral cavity.

Electrical characterisation of keratin with varying hydration shows a large decline in resistivity [25], which was reflected in our model by the systematic reduction in assigned resistivity values. When keratinised layer resistivity was reduced to 1 kΩm, expected trends with underlying tissue dysplasia (i.e., lower impedance for OPMD spectra) were recovered (Figure 7). It is difficult to justify reducing the keratinised layer resistivity any further as this would make it less resistive than the superficial cells of healthy keratinised tissue (~0.96 kΩm), which is unlikely given the increased keratin continent in the keratinised layer. Both the simulated (hydrated keratinised layer model) and measured low-frequency impedance (Figure 2) were approximately 4 kΩ for healthy tissue in this case. The agreement of the magnitude of simulated and measured low-frequency values gave confidence that the resistivity values for the keratinised layer in the simulations were reasonable. However, it was noted that the simulated impedance remained consistently high for frequencies below ~10 kHz, whereas in real measurements there was a gradual decrease in impedance from low to medium frequency instead of a plateau (see Figure 2). Assigning frequency-dependent properties to the keratinised layer based upon measurements of the stratum corneum [26] gave a more gradual reduction of impedance with frequency (see Figure 8). As with the previous study, keratinised layer resistivity was reduced to account for increased hydration in the mouth compared to the surface of skin. A reduction by a factor of 40 to ~1.2 kΩm (at 76 Hz) showed the best agreement with the measured data.

Following the fitting of frequency-dependent keratin layer properties, specific epithelial tissue structures representing all oral sites and pathologies listed in Table 1 were simulated. As shown in Figure 9, the keratinised (healthy), non-keratinised (severe OPMD) and para-keratinised (severe OPMD) spectra showed good agreement with measurements described previously [15] (Figure 2). While there is no breakdown of severe and moderate dysplasia in Murdoch et al.’s measurements, it is interesting that the moderate dysplasia (keratinised) case had a similar low-frequency impedance value to the healthy equivalent, but the impedance dropped off more rapidly with increasing frequency. Accounting for variation in layer depth derived from the histology analysis (represented by the error bars associated with each spectrum), aside from extremely low frequencies, the envelope of the predicted spectra for moderate dysplasia falls between the envelopes of the spectra obtained for healthy and severely dysplastic tissues with limited overlaps, suggesting that (providing the keratin properties are properly characterised) it may, in principle, be possible to identify early cancerous changes. The larger variation (see error bars, Figure 9) in impedance for OPMD and hyperkeratinous spectra can be explained by the greater variation in keratinous and superficial layer thicknesses (see Table 1).

One simulated spectrum did not fit well with the expected trend of OPMD tissues exhibiting lower transfer impedance at low frequency: the keratinised (severe OPMD) spectrum had a higher than expected low-frequency impedance (~5 kΩ). In addition, the predicted spectrum for non-dysplastic, hyperkeratinised tissue, which is a tissue type not previously explicitly represented in the measured data, appeared almost identical to that obtained from severely dysplastic keratinised tissue. These particular models have different epithelial tissue structures, due to difference in oral site (and in the OPMD example, pathological changes in epithelial structure) but have very similar keratinised layer thickness (86.61 μm and 88.85 μm for OPMD and hyperkeratinised tissue, respectively). These are the two thickest keratinised layers simulated in this work. A possible explanation for the similarity in these spectra could be that when the keratinised layer reaches a certain thickness, it prevents the current from reaching the epithelium and dominates the impedance measurements. In order to explore this further, a more detailed study of the effects of the keratinised layer in determining the spectra associated with normal and dysplastic tissue types was conducted, varying the keratinised layer thickness between 15 μm to 90 μm. As shown in Figure 10, spectra were sensitive to keratinised layer thickness with significant overlap between impedance values at the low-frequency end of the spectrum. Interestingly, for the range investigated, some separation between the two types was apparent in the mid-frequency range of 5–50 kHz, suggesting that this frequency range could be worthy of further investigation as the optimal choice for differentiating between keratinised normal, non-dysplastic hyperkeratinised tissue and dysplastic tissue structures.

Finally, given that oral epithelium can be underlain by either muscle, fat or bone, depending on the particular location within the mouth, we carried out a study to assess the likely influence on these sub-epithelial layers on the measured impedance spectra. This was explored by adapting our model to limit stroma depth, and the addition of underlying layers representing sub-stromal tissue, including fat (buccal tissue), bone (hard palate) and muscle (tongue). As is apparent from Figure 11, changes in predicted spectra obtained using these models for buccal tissue (cheek), hard palate and tongue were negligible, suggesting that the current flow through sub-stromal layers is insignificant in these cases, or that the electrical properties of the underlying layers were not significantly different to those above.

It is noted that in Figure 2 there are some spectra, collected from tissue subsequently identified as cancerous, that had a low-frequency impedance more typical of values recorded from healthy tissue. Given the similarity between the hyperkeratinised and OPMD spectra (see Figure 9), our results suggest that increased keratinisation of otherwise normal (non-dysplastic) epithelium may be responsible for some cases of misidentification. Although dysplastic epithelium is widely understood to be associated with lower impedance measurements, primarily as a result of the loss of cohesion between epithelial cells and the increase in extracellular space, this effect may be effectively ‘cancelled out’ by a relatively thick, moderately resistive keratinised layer, through which most of the current flow is contained. As suggested by a comparison of Figure 10 and Figure 11, this surface layer has a significantly larger effect on measured impedance than the presence of sub-stromal layers at different sites in the oral cavity, such as fat, bone or muscle.

The predicted link between relatively high impedance and low frequency and keratinisation of tissues has implications for exactly how EIS might be applied in order to identify OPMD in the mouth. It is certainly a risk that focusing on low-frequency impedance readings (where the potential range of measured values is much larger than the high-frequency case, where most spectra converge) may potentially lead to keratinised OPMD tissue being misdiagnosed as normal. However, given that OPMD occurs primarily at sites that are not normally keratinised (lateral tongue, floor of the mouth) and OPMD progression is associated with keratinisation, it is possible that the diagnostic potential of EIS lies in detecting locations where the impedance measured at low frequency is actually higher than normal. In order to explore this, we would need to extend this study to simulate a range of tissue pathologies associated with the same oral sites. Likewise, in vivo measurement studies would ideally be repeated, with subsequent biopsies used to identify keratinisation status, in addition to stage of dysplasia.

Several factors contributing to uncertainties remain to be fully quantified: as explored in our initial studies, there is considerable uncertainty associated with the electrical properties of any keratinised layer in the oral cavity. As set out above, we have carried out a systematic study of the effect of properties, ranging from frequency, independent, highly resistive dry keratin, frequency-dependent stratum corneum-based properties, and finally allowing for the hydrated state of keratin, as would be expected in a moist environment such as the mouth. Although this was effectively a fitting study, our assumptions are justified at least qualitatively, and allowed the subsequent exploration of the relative effect of other factors. More precise measurement of the properties associated with keratin in the oral environment would further improve the accuracy of our model.

In a previous FEM study of the cervix [20], the presence of a thin conductive surface mucus layer was found to be key to producing realistic simulations. The analogue to the mucus layer in this work is a saliva layer between the electrodes and underlying tissue. Reliable material properties were not available in the literature; however, it is reasonable to assume that the resistivity of saliva is likely to be far less than the underlying tissue layers. The thickness of the saliva layer is not explored in this particular study, but it is reasonable to assume that it would remain constant throughout the oral cavity and for normal and dysplastic tissue types. Although more detailed elucidation of its thickness and electrical properties would improve the realism of the model, it is not expected to have a significant impact on the ability to differentiate underlying tissue types. Likewise, our model simplifies the basal surface of the stroma as being flat and parallel to the tissue surface, neglecting to include the rete pegs, which are a distinguishing feature of several oral tissue sites. Our results suggest that the current flow is confined to the surface, epithelial and stromal layers and thus the presence of such structures is unlikely to have a significant effect on the measured spectra. However, future work is planned to explore this further.

This work focused on generating models of oral tissue structures based on the morphological data available in histology images. Previous work on the cervix has demonstrated the distribution of extracellular space (ECS) with depth in epithelium is an important factor in determining the low-frequency impedance characteristics of normal and dysplastic tissue. Due to the lack of ultra-high-resolution microscopy data specific for oral tissue, an assumption in this work is that the ECS thickness values from cervical precancer can be applied to oral tissue. Given the similarities between the squamous epithelia of the cervix and oral cavity, the values used are likely to be of the correct order, given the tissue scale results’ agreement with real measurements, but the exact ECS thickness remains a significant source of uncertainty for the cellular scale simulations. The same can be said for the electrical properties of the cellular components, which are averaged from the values in the literature [16] (a full breakdown of references is given in Appendix A). However, a sensitivity study of electrical and morphological parameters of cervix cells showed ECS thickness to be more significant [20]. In the event of further parameter data (a full list is given in the conclusions) relating specifically to oral tissues becoming available, this can be incorporated into our model.

## 5. Conclusions

A multiscale modelling approach has been applied to the simulation of EIS for oral tissue based upon real histology data. Electrical properties were sourced from the literature when available or fitted to real EIS measurements [15] where not. Key findings include:Simulations suggest differences between spectra obtained from healthy and non-keratinous OPMD tissue, with good agreement with the low-frequency impedance and general form of real measurements.The exception is keratinised OPMD tissue, where higher than expected impedance values are predicted.Given the overlap of predicted spectra for keratinised and non-keratinised severe and moderate dysplasia (OPMD), it is unlikely that EIS will provide a reliable method of differentiating between these tissue types.Our results suggest that high impedance readings obtained from OPMD tissue can be caused by increased keratinisation. Measurements at frequencies in the range 5–50 kHz may provide the optimum range to differentiate between tissue types.The presence of different sub-stromal underlying tissue types according to oral site is unlikely to have a significant impact on the measured impedance spectra.

Parameters that are currently a significant source of uncertainty in the models:Extracellular space thickness in oral tissue and how it varies with dysplasia.Keratinised layer electrical properties.Electrical properties and morphology of the saliva layer.

Obtaining real measurements of these parameters would greatly aid the modelling and hence future interpretation of measured EIS spectra.

Future explorations with our model will focus on optimising the size and separation of arrangements for the surface tetrapolar array in order to suggest configurations that are likely to be most effective in terms of differentiating healthy and dysplastic tissue, and potentially moderate and severe grades of dysplasia.

## Figures and Tables

**Figure 1 sensors-22-05913-f001:**
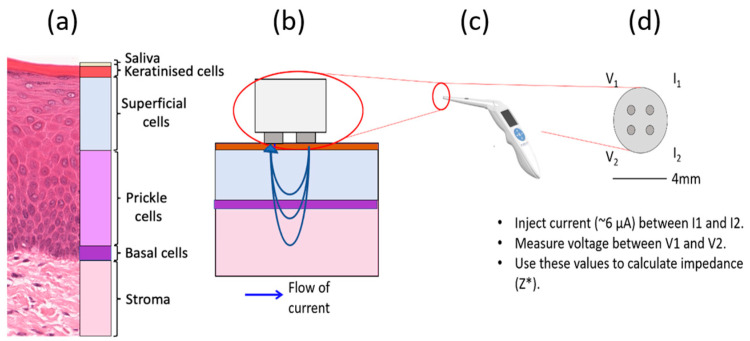
Schematic of a four-point probe EIS measurement of oral tissue. (**a**) Histology image of normal hard palate with cellular compartments annotated. (**b**) The probe tip in contact with oral tissue and the flow of current, (**c**) the ZedScan handset and (**d**) the tetrapolar probe tip with labelled electrodes.

**Figure 2 sensors-22-05913-f002:**
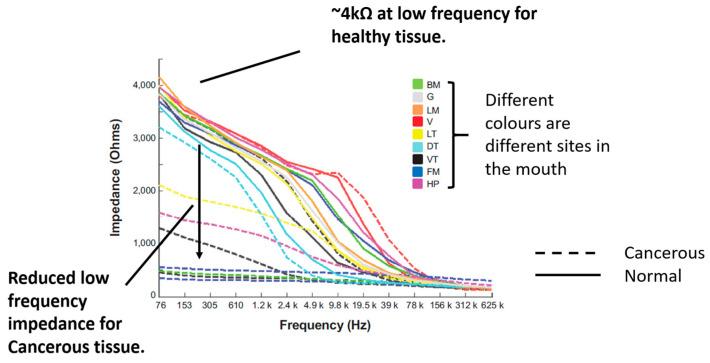
Measured impedance spectra of various oral sites and diagnoses. Adapted from [15] (*International Journal of Nanomedicine* 2014 9 4521–4532. Originally published by and used with permission from Dove Medical Press Ltd.). Note the severe OPMD cases in the present study are the closest match to confirmed cancerous tissue considered by the previous work above.

**Figure 3 sensors-22-05913-f003:**
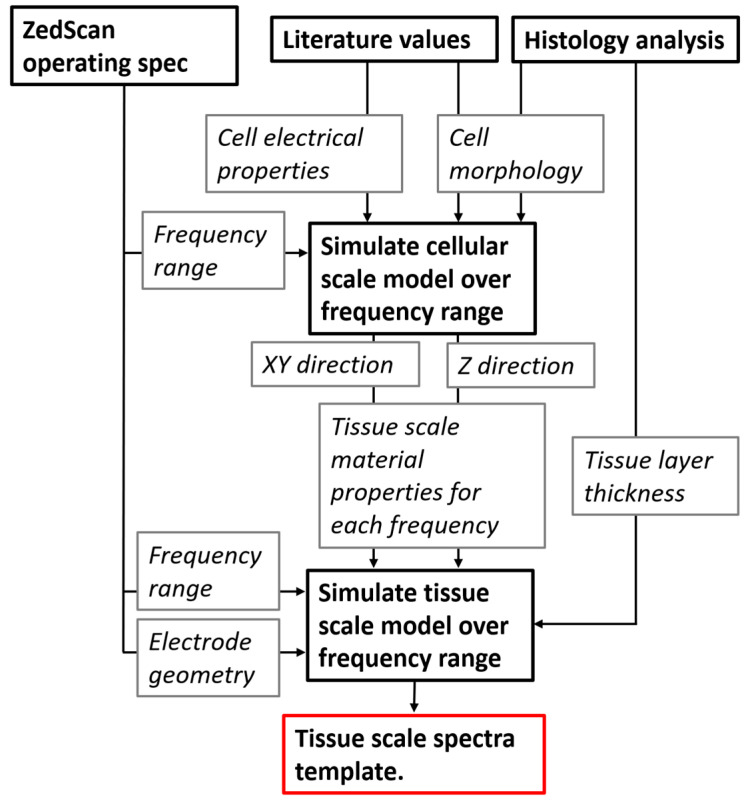
Flow of information between the cellular and tissue scale models including inputs/constraints (current frequency and electrode geometry) associated with the ZedScan system.

**Figure 4 sensors-22-05913-f004:**
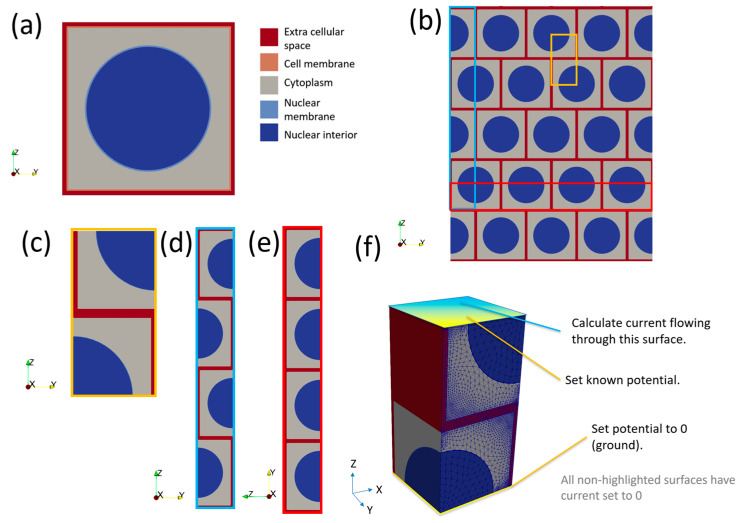
(**a**) Cross-section of a generic (nucleated) cell model of length 5 μm. Note the cell and nuclear membranes have been exaggerated to make them visible. (**b**) Bricked layer representation of cells; note the pattern is assumed to repeat infinitely in the X direction (in/out of the page). (**c**) The smallest repeating unit of (**b**) in 2D. (**d**) The Z direction arrangement of cells. (**e**) The XY direction arrangement of cells. (**f**) The smallest repeating unit of (**b**) in 3D with boundary conditions applied to cellular scale model to calculate impedance.

**Figure 5 sensors-22-05913-f005:**
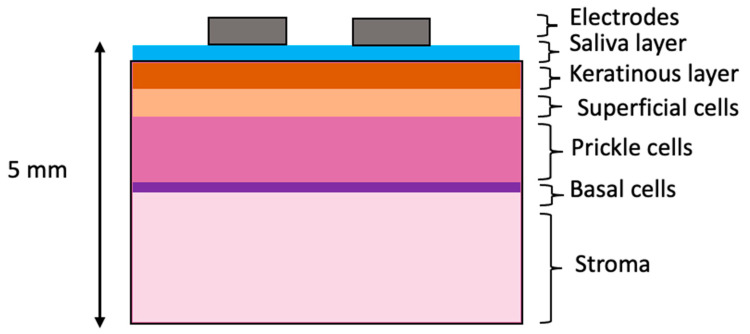
Schematic of tissue-level layered model. Note that layer thicknesses are not to scale.

**Figure 6 sensors-22-05913-f006:**
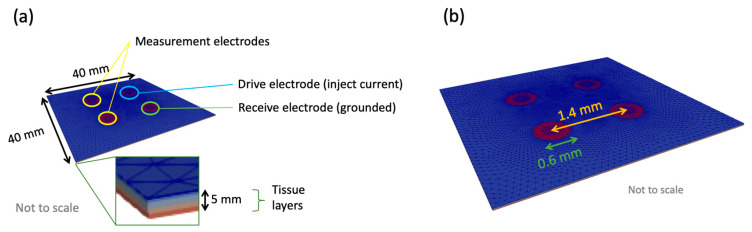
(**a**) A tissue scale model with dimensions and electrodes indicated. Potential difference was calculated between the two measurement electrodes (defined as coupled sets in Ansys). (**b**) Electrode dimensions for the tissue scale model.

**Figure 7 sensors-22-05913-f007:**
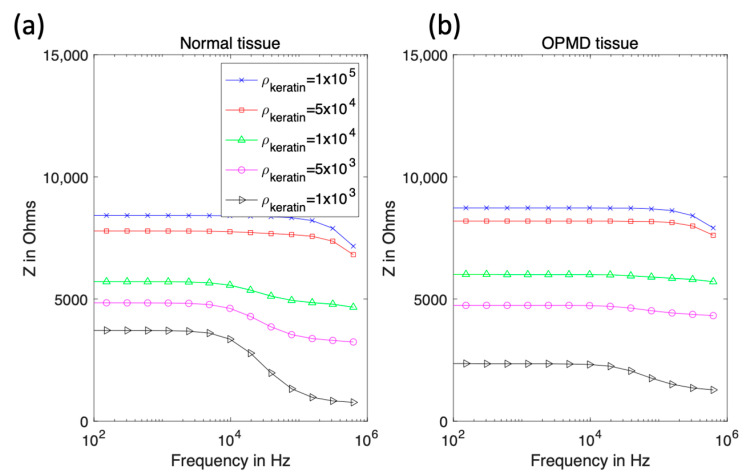
Real impedance spectra for keratinised normal (**a**) and OPMD (**b**) tissue with varying keratinised layer resistivity.

**Figure 8 sensors-22-05913-f008:**
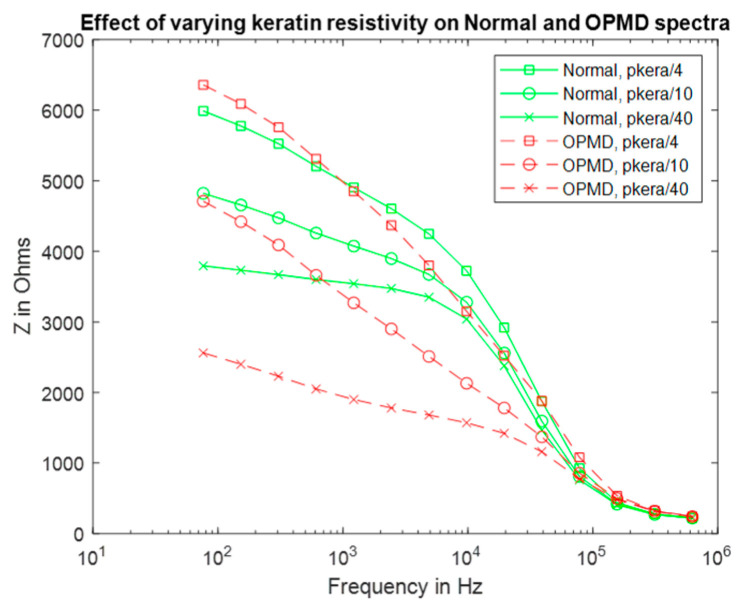
Real impedance spectra for normal and OPMD oral tissue according to keratinised layer resistivities (pkera).

**Figure 9 sensors-22-05913-f009:**
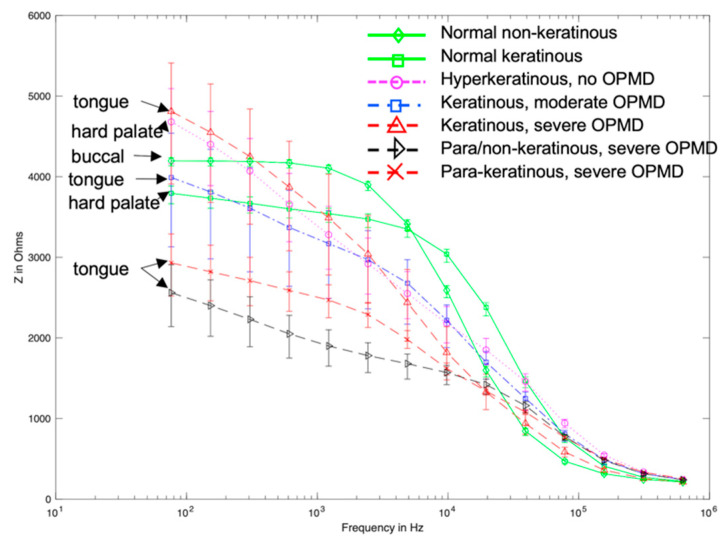
Modelled spectra templates for oral sites of varying keratinisation and pathology. Error bars represent the range of spectra obtained when simulating maximum and minimum measured cellular compartment thickness for each tissue type.

**Figure 10 sensors-22-05913-f010:**
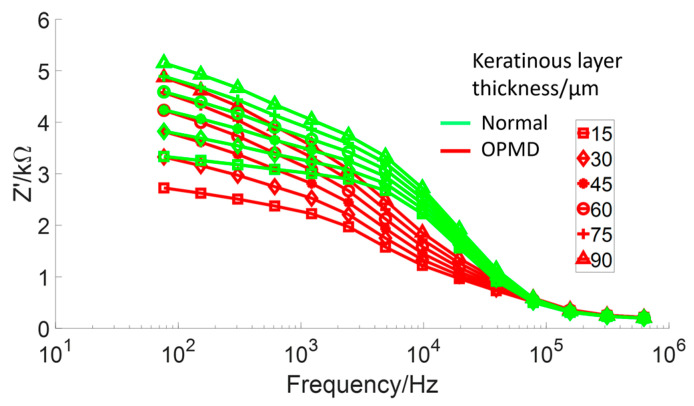
Comparison of healthy and severely dysplastic oral tissue impedance spectra with increasing keratinised thickness.

**Figure 11 sensors-22-05913-f011:**
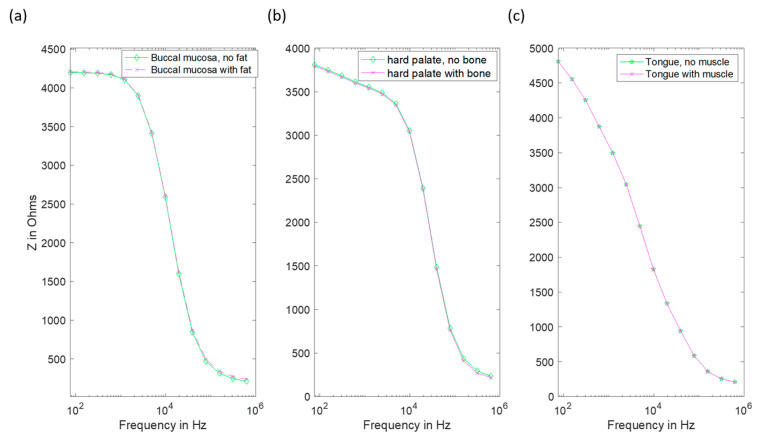
Impedance spectra of several oral sites with and without explicit inclusion of the appropriate underlying layer. (**a**) The buccal mucosa with added fat. (**b**) Hard palate with added bone. (**c**) Tongue with added muscle layer (note the coincident data points in this case).

**Table 1 sensors-22-05913-t001:** List of all measurements used in the tissue scale models, as obtained from histological analysis or oral tissues. Moderate and severe dysplasia are forms of OPMD.

		Tissue Type (Normal for Site)		Mean Tissue Layer Thickness/μm			
Case	Diagnosis		Oral Site	Basal	Prickle	Superficial	Keratinised
1	Normal	Non-keratinised	Buccal mucosa	12.4 ± 4.5	150.4 ± 28.8	274.9 ± 46.3	n/a
2	Normal	Orthokeratinised	Hard palate	15.7 ± 2.9	100.5 ± 8.5	182.0 ± 45.8	14.0 ± 1.8
3	Severe dysplasia	Parakeratinised	ventral tongue	22.1 ± 4.6	73.9 ± 13.5	261.7 ± 58.2	86.6 ± 34.0
4	Severe dysplasia	Parakeratinised	lateral tongue	25.4 ± 4.6	225.5 ± 36.6	289.5 ± 102.9	22.8 ± 7.0
5	Moderate dysplasia	Parakeratinised	lateral tongue	15.6 ± 4.1	168.0 ± 83.8	287.3 ± 102.9	42.9 ± 17.9
6	Hyper-keratosis No dysplasia	Orthokeratinised	Hard palate	15.1 ± 3.4	222.1 ± 16.2	363.3 ± 85.8	88.9 ± 21.5
7	Severe dysplasia	Parakeratinised	lateral tongue	28.0 ± 6.8	387.7 ± 53.8	356.9 ± 68.3	17.3 ± 8.4

## Data Availability

The data presented in this study are available in the Appendix A.

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
