# Peer review of "Computational Modelling for Electrical Impedance Spectroscopy-Based Diagnosis of Oral Potential Malignant Disorders (OPMD)"

_sensors, 2022, doi:10.3390/s22155913_

Round 1

Reviewer 1 Report

This manuscript seems good and interesting. This is well written. The manuscript may be considered for publication.

Reviewer 2 Report

In this paper a multiscale finite element modelling approach is employed to explore the factors underlying the differences in characteristic spectra associated with different oral tissue types. The authors are interesed to extend computational modelling methodologies (previously applied to other epithelial tissues), in order to fully understand the electrical characteristics of the much more varied (in terms of thickness and keratinisation) tissues found in the mouth. The considered the long-term goal is that this knowledge will aid the interpretation of electrical impedance spectroscopy (EIS) spectra and inform the feasibility of applying this measurement technique for the discrimination of oral tissue pathologies.

The work  involves Computational Modelling work focussed on generating models of oral tissue structures based on the  morphological data available in histology images

I have a question:
The authors wrote: "In the event of further parameter data relating specifically to oral tissues becoming available, this can be incorporated into our model". What would those parameters be?

The manuscript is interesting, well wirtten and easy to read.

Reviewer 3 Report

This work is intersting. I recommend acepting it.

Reviewer 4 Report

This paper from Walker and coworkers details a multiscale modelling approach to simulate the electrical properties of oral tissue, providing a method to diagnose oral potential malignant disorder. In particular, the simulations provide an insight to distinguish moderate dysplasia from severe dysplasia or healthy tissue. Even though the OPMD is possible to be misidentified in the simulation, the author proposed remedies. The manuscript is well organized. Especially, the background part provides enough information to help reader understand the history and necessity of the research. I believe the work is well suited for Sensors. A couple minor points for consideration on revision are listed below.

1   1. The authors use the same type of tissue to do different measurements. Are these tissues fresh in different measurements or reused?

2   2.The authors mention that the hydration is able to reduce resistivity of tissues in the measurements. I would like to question why the hydration is able to reduce the resistivity. Is this due to the ions or some other reasons? If it is ions, if the accuracy of simulations can be optimized by changing the type and concentration of ions.

3   3. Is it on purpose to capitalize the word “Its” (page 7, line 240); From my monitor, in figure 9, the color of line is not in consistence with the labels in legend, please double check it.   
